# Topical Reappraisal of Molecular Pharmacological Approaches to Endothelial Dysfunction in Diabetes Mellitus Angiopathy

Constantin Munteanu [1,2,*], Mariana Rotariu [1], Marius-Alexandru Turnea [1], Aurelian Anghelescu [2,3], Irina Albadi [4,5], Gabriela Dogaru [6,7], Sînziana Calina Silișteanu [8], Elena Valentina Ionescu [4,9], Florentina Carmen Firan [10], Anca Mirela Ionescu [11], Carmen Oprea [4,9] and Gelu Onose [2,11,*]

1. Faculty of Medical Bioengineering, University of Medicine and Pharmacy "Grigore T. Popa" Iași, 700454 Iași, Romania; mariana.rotariu@umfiasi.ro (M.R.); marius.turnea@umfiasi.ro (M.-A.T.)
2. Neuromuscular Rehabilitation Division, Teaching Emergency Hospital "Bagdasar-Arseni", 041915 Bucharest, Romania; aurelian.anghelescu@umfcd.ro
3. Faculty of Midwives and Nursing, University of Medicine and Pharmacy "Carol Davila", 020022 Bucharest, Romania
4. Faculty of Medicine, Ovidius University of Constanta, 900470 Constanta, Romania; irina.albadi@yahoo.com (I.A.); elena.ionescu@365.univ-ovidius.ro (E.V.I.); carmen.oprea@365.univ-ovidius.ro (C.O.)
5. Teaching Emergency County Hospital "Sf. Apostol Andrei", 900591 Constanta, Romania
6. Faculty of Medicine, "Iuliu Hatieganu" University of Medicine and Pharmacy, 400012 Cluj-Napoca, Romania; gabriela.dogaru@umfcluj.ro
7. Clinical Rehabilitation Hospital, 400437 Cluj-Napoca, Romania
8. Faculty of Medicine and Biological Sciences, "Stefan cel Mare" University of Suceava, 720229 Suceava, Romania; sinziana.silisteanu@usm.ro
9. Balneal and Rehabilitation Sanatorium of Techirghiol, 906100 Techirghiol, Romania
10. Teaching Emergency Hospital of the Ilfov County, 022113 Bucharest, Romania; firancarmen@yahoo.com
11. Faculty of Medicine, University of Medicine and Pharmacy "Carol Davila", 020022 Bucharest, Romania; anca.ionescu@umfcd.ro
* Correspondence: constantin.munteanu.biolog@umfiasi.ro (C.M.); gelu.onose@umfcd.ro (G.O.)

**Abstract:** Diabetes mellitus (DM) is a frequent medical problem, affecting more than 4% of the population in most countries. In the context of diabetes, the vascular endothelium can play a crucial pathophysiological role. If a healthy endothelium—which is a dynamic endocrine organ with autocrine and paracrine activity—regulates vascular tone and permeability and assures a proper balance between coagulation and fibrinolysis, and vasodilation and vasoconstriction, then, in contrast, a dysfunctional endothelium has received increasing attention as a potential contributor to the pathogenesis of vascular disease in diabetes. Hyperglycemia is indicated to be the major causative factor in the development of endothelial dysfunction. Furthermore, many shreds of evidence suggest that the progression of insulin resistance in type 2 diabetes is parallel to the advancement of endothelial dysfunction in atherosclerosis. To present the state-of-the-art data regarding endothelial dysfunction in diabetic micro- and macroangiopathy, we constructed this literature review based on the Preferred Reporting Items for Systematic Reviews and Meta-Analyses (PRISMA). We interrogated five medical databases: Elsevier, PubMed, PMC, PEDro, and ISI Web of Science.

**Keywords:** diabetes mellitus; endothelial dysfunction; microangiopathy; macroangiopathy

## 1. Introduction

Diabetes mellitus (DM) is a metabolic disorder of multiple etiology characterized by chronic hyperglycemia [1]. Micro- and macrovascular complications that develop in DM patients are a consequence of numerous factors, including, most importantly, endothelial dysfunction [2]. In normal physiological status, there is a suitable equilibrium of relaxing and contractile elements released from the endothelium. However, this delicate state is

altered in diabetes, thus contributing to the subsequent progression of vascular lesions and affected organs. Therefore, the morbidity and mortality encountered in DM are mainly determined by its vascular complications, represented by micro- and macroangiopathy [3,4] and reflecting the dysfunction of blood vessels [5].

Microangiopathy is characterized by pathogenesis at the microvascular level, involving capillaries, as is the case with diabetic retinopathy, which can cause blindness; diabetic nephropathy (resulting in renal failure); and neuropathy [6,7], described as vasa nervorum angiopathy [8].

The hormonal and physiological dysfunctions associated with diabetes, including oxidative stress, insulin resistance, reactive oxygen production, advanced glycation end products (AGEs), and increased inflammatory cytokine levels, jointly promote atherosclerotic cardiovascular disease [9]. Atherosclerosis is the main form of macroangiopathy in DM [3], affecting larger blood vessels supplying the brain, heart, and extremities.

Peripheral vascular disease is a frequent complication in type 2 DM, which has, as its action mechanisms, atherosclerosis, endothelial dysfunction, vascular remodeling, and thrombosis [2].

According to various research hypotheses, the adverse effects of diabetes on the vascular system through the influence of hyperglycemia [10], especially on the endothelium, as possible targets include changes in cellular redox status, altered oxidized/reduced forms of nicotinamide adenine dinucleotide (NAD+/NADH) ratio [11]; dysregulation of tyrosine kinase proteins, whose activity is influenced by the redox state of their sulfhydryl groups [12]; protein kinase C (PKC) disorder [13]; and sorbitol accumulation [3]. In addition to being a vital part of the body's metabolic environment, insulin also has other physiological functions, such as regulating the blood flow and increasing glucose uptake in skeletal muscle. The actions of insulin-stimulating agents, by stimulating the production of vascular endothelium's nitric oxide (NO), can also improve the blood flow and increase glucose uptake in skeletal muscle. The actions of insulin are mainly mediated by the phosphatidylinositol-3-kinase (PI3K) [14].

This paper aims to underline the molecular mechanisms of endothelial dysfunction in DM microangiopathy and macroangiopathy and further our understanding of the pharmacological molecular approaches oriented to improving the dysfunctional status of the vascular endothelium and DM evolution control.

## 2. Methods and Results

This synthetic and systematic literature review is based on the PRISMA methodology, by searching free full-text available papers written in English, appearing in the last five years (1 January 2017–31 December 2021), by a specific keyword combination: "diabetes AND angiopathy", in well-known international databases: Elsevier, National Center for Biotechnology Information (NCBI)/PubMed, NCBI/PubMed Central (PMC), Physiotherapy Evidence Database (PEDro) and—to verify whether the works found were published in ISI (Institute for Scientific Information—ex Thomson Reuters—currently administered by Clarivate Analytics) -indexed journals—the Web of Science. All of the literature and clinical trial databases mentioned above were searched from January to April 2022. The inclusion criteria were fixed regarding patients with diabetes (as diagnosed using recognized diagnostic criteria). Exclusion criteria correspond to study dates not before 2017. After this first step, in the second step, we removed duplicates (same work found in two or more databases). In the third step, we checked for and retained only those issued in ISI-indexed publications. The scientific impact of each article was established using a customized quantification formula based first on a PEDro-adapted quantitative method, considering the number of citations per year, all of the values being limited between 0 and 10, using a mathematical equation. Only articles that reached at least 4 points were selected. Works that obtained a score of at least 4 were considered eligible ("fair quality = PEDro score 4–5").

*Search and Filtering Results*

Database interrogation provided, initially, 207 articles. Applying the PRISMA selection filters and scoring resulted in 29 unique published qualified studies (Figure 1). The PRISMA-standardized methodology for conducting systematic reviews entails/requires specific steps and a high level of strictness. Such a systematic process is meant to help the authors construct an article structure skeleton giving a primary objective input about assembling the literature background to be approached, summarized, and synthesized. Moreover, the afferent contextual search (by keywords combination/syntaxes) we fulfilled may sometimes reduce the number of obtained articles. The 29 selected papers helped us complete our first task of constructing our article structure skeleton.

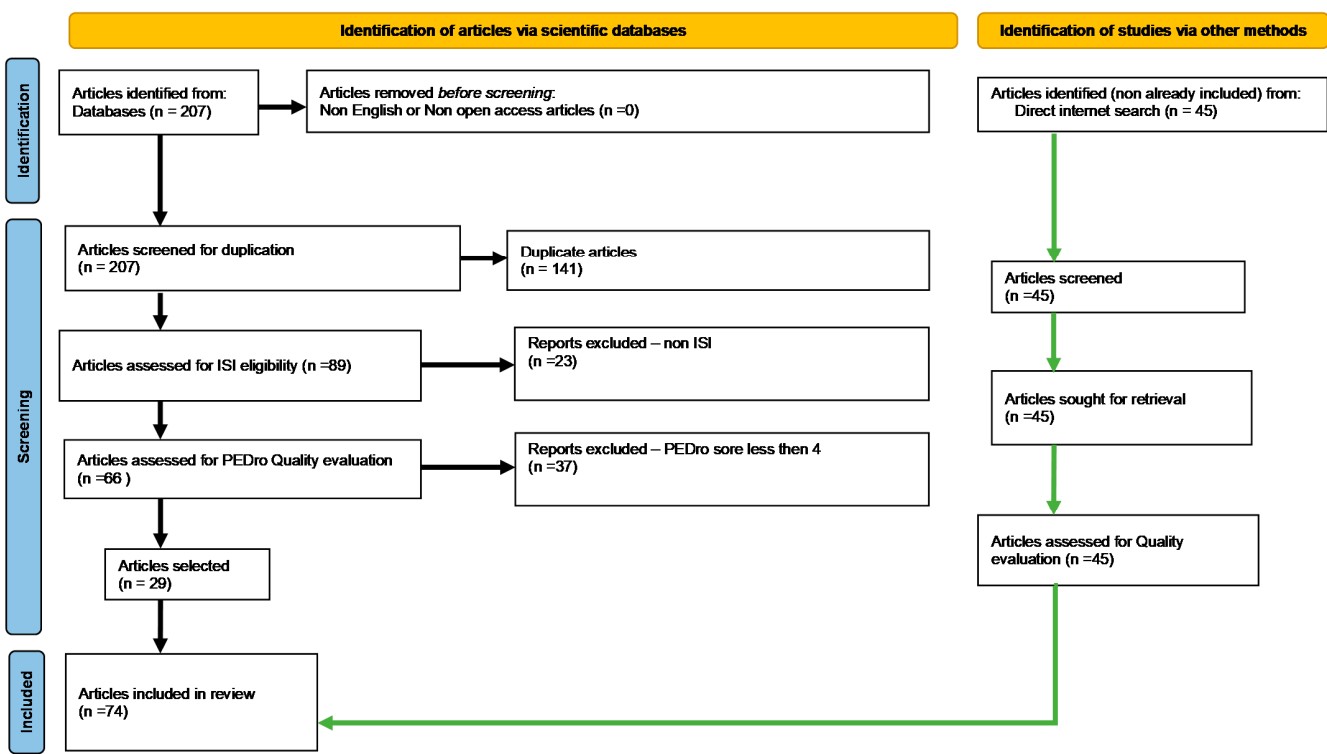

**Figure 1.** PRISMA-type flow diagram for searching free full-text available papers (ISI eligibility–articles published in ISI-Web-of-Science-indexed journals).

Of course, although rigorous and broadly recognized and used, the PRISMA filtering methodology is not infallible, so there are always possible limitations in covering all the necessary knowledge. In the meantime, to fill the knowledge gaps, we enhanced our documentation with other bibliographic resources connected to our subject, identified in the literature, within a non-standardized search.

### 3. Diabetes Mellitus—Vascular Pathological Interplay Picture

DM is a frequent chronic metabolic pathology that causes disorders in the metabolism of carbohydrates, electrolytes, fats, and proteins on a pandemic scale [15], affecting over 400 million people worldwide and being estimated to affect over 650 million people over the next 25 years. DM can also be defined as a syndrome associated with chronic hyperglycemia, resulting from a reduced insulin secretion from the pancreas or a lack of the effect of insulin. Factors such as lower insulin in the blood, reduced blood glucose consumption, or increased blood glucose production, may result in high blood glucose levels [16].

The criteria for glucometabolic disturbances, as established by the World Health Organization (WHO) and the American Diabetes Association (ADA) [1], include etiological types and different clinical stages of hyperglycemia [1]. Four main etiological categories have been identified as diabetes type 1, type 2, other specific types, and gestational diabetes:

- Type 1 or juvenile diabetes is characterized by a deficiency of insulin secretion caused by a cellular-mediated autoimmune destructive lesion of pancreatic β-cells [17] after exposure to viral, toxic, or other environmental triggers [18]. This type has a genetic susceptibility or an immune disorder causality and accounts for about 10% of DM cases. In type 1 diabetes, endothelial dysfunction is a key determinant of inflammatory activity and is considered an early marker of cardiovascular diseases (CVD) [19]. The inflammatory response is generated by altered immunity, including increased cytokine and chemokine, superoxide release, and leukocyte marginalization [8]. Autoantibodies known to be involved in the development of type 1 diabetes may help identify individuals with this condition, but they do not have sufficient diagnostic accuracy. For example, although low levels of C-peptides in the blood are known to help identify individuals with type 1 diabetes, they do not help treat acute hyperglycemia [20].

- Type 2 is a heterogeneous disorder [21], characterized by a relative deficiency of insulin secretion [22], also seen as a primary event in disease development [21], and peripheral tissue targeting insulin resistance. A decreased number of insulin receptors in the target cells is correlated with a lack of insulin response, impaired endothelial signal transduction, and regulated redox activation of transcription factors [15]. Endothelial dysfunction in type 2 diabetes has also been shown to appear. In addition, excessive production of reactive oxygen species (ROS) due to hyperglycemia has been shown to induce epigenetic changes, such as histone 3 lysine monomethylation, which increases the expression of the p65 subunit of NF-κB. These epigenetic reactions can be mediators between diabetes, chronic inflammatory response, and cardiovascular disease (CVD) [10]. Around 80% of people with type 2 diabetes develop cardiovascular complications. This group accounts for over 70% of deaths in this population. The DISCOVER study [23] provided a comprehensive view of the various aspects of cardiovascular disease in people with diabetes after using a standardized case report form, which allows for comparisons of the results across different regions and countries. The study included 15,992 individuals with diabetes [23].

Overnutrition and chronological age are the main risk factors for developing type 2 diabetes mellitus (T2DM). These two conditions promote a chronic state of inflammation, or low-grade inflammation (LGI), which is referred to as a phenomenon termed either "inflammaging" or "metaflammation". Despite being chronic, this condition can still lead to the development of acute inflammatory responses [24].

The four categories of vascular complications include: macrovascular, microvascular, cerebrovascular, and peripheral artery disease. Macrovascular disease is a type of vascular disease that can affect the central and peripheral arteries. It can also lead to various conditions, such as heart failure and amputation. Microvascular disease is a type of vascular disease that can affect the small blood vessels in multiple organs in the body [23].

The prevalence of microvascular disease is 18.8% globally, with the lowest in Africa at 14.5%. Some of these include: peripheral neuropathy, 7.7%; chronic kidney disease, 5%; albuminuria, 4.3%; retinopathy, 2.7%; erectile dysfunction, 1.0%; and retinal laser photocoagulation, 0.6%. The most common type of vascular disease was peripheral neuropathy, the most prevalent type of vascular disease in all regions except for the Western Pacific. After taking into account the sex and age of the patients, the prevalence of microvascular disease was 17.9%. It was highest in Europe (20.4%) and lowest in the Americas (14.2%) [23].

The prevalence of vascular complications was 12.7%. These include: coronary artery disease, 3.3%; heart failure, 2.2%; stroke, 1.2%; peripheral artery disease, 0.7%; and transient ischemic attack, 0.1%. The most common type of vascular disease was coronary artery disease. The findings show that early and aggressive risk factors can help prevent vascular complications [23].

In addition to these two main types of DM, the World Health Organization (WHO) has begun to classify particular types of DM, including malnutrition-related diabetes (that accompanies certain conditions and syndromes) and gestational diabetes. The most common of these particular types is the gestational one, which is similar to type 2 DM and

is also related to insulin resistance, which develops in pregnant women due to pregnancy hormone impregnation [25].

Acute, life-threatening consequences of uncontrolled diabetes are hyperglycemia with ketoacidosis or nonketotic hyperosmolar syndrome. Pronounced hyperglycemia manifestations include polyuria, polydipsia, weight loss, sometimes polyphagia, and blurred vision [26]. In addition, persistently high blood sugar causes microvascular damage, including nephropathy, retinopathy, or neuropathy. Macrovascular complications include coronary heart disease, leading to myocardial infarction, cerebrovascular disease, stroke [27], and peripheral vascular disease. The pathogenesis of vascular complications is multifactorial, caused by glucose-mediated endothelial damage, oxidative stress, sorbitol production, and advanced glycation end products (AGE) [28].

The microvascular complications of DM are essentially diseases of the small blood vessels, while macrovascular complications are associated with wide blood vessels. Diabetic nephropathy, retinopathy, and neuropathy, which can cause terminal renal disease, blindness, and sharp neuropathic pain, can be described as altered microvascularity that affects the capillary basement membrane entailing the arterioles of the retina, glomeruli, myocardium, muscle, and skin vessels. Macroangiopathy refers to damage to the arterial walls of the peripheral and coronary artery systems, but such macrovascular complications occur mainly due to kidney disease in DM. These include cerebrovascular, cardiovascular, and peripheral vascular problems. Under hyperglycemia, microangiopathy and macrovascular complications arise due to complexities and structural or functional anomalies, such as hypertrophy, inadequate blood flow, and loss of specific functions [11].

In obesity, visceral fat deposition determines inflammation processes, which are essential in DM complications. In addition, the association between high blood pressure and obesity is also known to cause a higher rate of morbidity and mortality associated with CVD. About 10–30% of type 1 diabetics and 60% of type 2 diabetics are reported to suffer from high blood pressure [29].

DM's pathogenesis and its complications involve many genetic constituents and biochemical pathways changing the metabolic fingerprint. It affects the cellular transcripts of target organs and biochemical pathways that result in abnormal gene expression of pro-apoptotic, pro-fibrotic, pro-inflammatory, and growth-promoting genes [30].

Epigenetics plays an essential role in gene interactions. The major molecular events tangled in the epigenetic phenomenon are histone changes, DNA methylation, and microRNA interference. For example, research studies reported that disruption of signaling pathways (inflammation, apoptosis, and stress due to increased oxidation, etc.) occurs due to abnormal changes in histones, DNA methylation, and modulated expression of miRNAs and lncRNAs in T2DM [30].

## 4. Endothelial Dysfunction in Diabetes Mellitus

The endothelium is a crucial part of the cardiovascular system, which orchestrates many aspects of microvascular function, including permeability, pressure, flow, angiogenesis, and rheology [31]. The function of the vascular endothelium is altered in DM, conducting the clinical expression of microangiopathies. Diabetic endothelial dysfunction is thought to be generated by an early increase in vascular permeability [32]. Under physiological conditions, there is a balanced release of relaxing and contractile factors derived from the vascular endothelium. However, this delicate balance is altered in diabetes, thus contributing to the progression of vascular lesions. Diabetic endothelial dysfunction is thought to be manifested by an early increase in vascular permeability. The fiber-matrix theory of microvascular permeability holds that the ability of the capillary wall to act as a molecular sieve is a feature conferred by the endothelial glycocalyx [33].

Endothelial cells (ECs) actively regulate permeability, vascular tone, coagulation, and fibrinolysis factor equilibrium, the composition of the subendothelial matrix, the proliferation of vascular smooth muscle cells, and extravasation of leukocytes [3]. To fulfill these functions, the endothelium produces extracellular matrix components and a large

spectrum of regulatory mediators, such as NO, endothelin, prostanoids, angiotensin II, tissue-type plasminogen activator t-PA, and plasminogen-1 activator inhibitor (PAI-1), von Willebrand factor (vWF) [34], adhesion molecules, and cytokines. For example, the endothelium usually decreases vascular tone, a process in which nitric oxide plays a key role. In addition, it regulates vascular permeability to nutrients, hormones, and other macromolecules and leukocytes. Endothelial cells (EC) normally inhibit platelet adhesion and aggregation by producing prostacyclin, nitric oxide, and ectonucleotidases, limiting the activation of the coagulation cascade by thrombomodulin–protein C and heparan sulfate–antithrombin III interactions, and regulate PA t-production [11].

The endothelium responds to specific local and temporal (Figure 2) needs by releasing NO or t-PA after stimulation and induction of genes, for example, E-selectin, after exposure to inflammatory mediators. Endothelial dysfunction can be expressed by changes in cell-matrix interactions, increased vascular permeability, high vascular tone, and the loss of antithrombotic and profibrinolytic properties, and may, in turn, acquire prothrombotic and antifibrinolytic properties [35]. However, such changes do not necessarily coincide. In addition, they may differ depending on the nature of the lesion and may relate to the intrinsic properties of the endothelium (for example, venous versus arterial versus microvascular endothelium) [3].

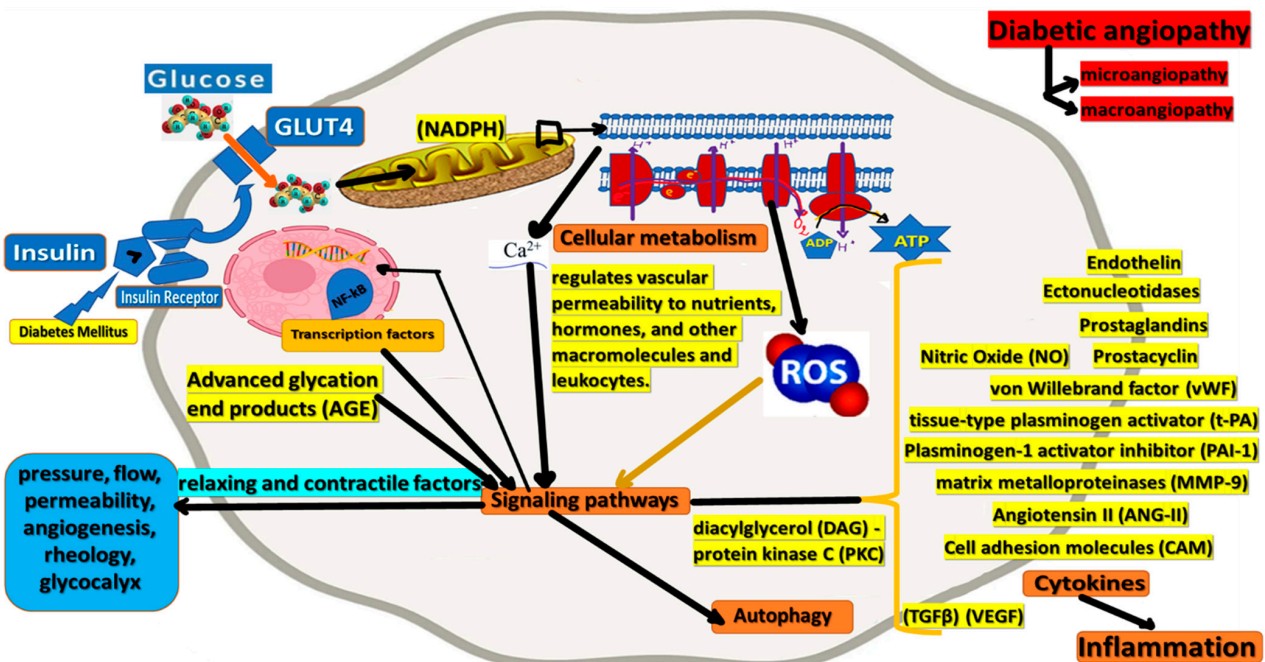

**Figure 2.** Endothelial cells (ECs) fulfill specific physiologic functions such as actively regulating permeability, vascular tone, coagulation, and fibrinolysis factor equilibrium, the composition of the subendothelial matrix, and the proliferation of vascular smooth muscle cells, and cytokine release. Dysfunctional endothelium, mainly due to high levels of ROS, is a potential contributor to the pathogenesis of vascular disease in diabetes.

Vascular endothelial cells activated by hyperglycemia and/or AGE express cell adhesion molecules (CAM) by stimulating cytokines, tumor necrosis factor-alpha (TNFα), and interleukin-1 (IL-1). The intercellular cell adhesion molecule (ICAM-1) is a marker of white blood cell (WBC) interaction with the vascular endothelium and facilitates the transendothelial migration of blood cells into the subendothelial space. Von Willebrand factor (vWF) is considered a more conventional marker of endothelial dysfunction. The mechanisms of such endothelial dysfunction are still unclear, but free radical generation and oxidative stress have been proposed as key mechanisms of vascular injury in diabetes [19].

Metabolic disorder in diabetes is associated with excess ROS generation through polyol, glucose self-oxidation, prostanoid synthesis, and protein glycosylation [19]. An additional mechanism that increases ROS generation via activation of polyamine metabolism, involved in cell survival, cell proliferation, and protein synthesis, contributes to diabetes pathogenesis under inflammation conditions [36]. Many studies in animal models and human clinical trials strongly suggest that hyperglycemia is a crucial feature of diabetic microvascular complications [32].

Consistent data support that free fatty acids are responsible for the association between insulin resistance and visceral adiposity [25]. In addition, growing evidence underpin that the progression of insulin resistance in type 2 diabetes is parallel to the advancement of endothelial dysfunction in atherosclerosis, and hyperglycemia can be considered the primary factitive factor linked with the development of endothelial dysfunction [37].

The production of advanced glycation end products (AGEs) is an essential multifactorial biochemical abnormality that accompanies diabetes and (probably) inflammation. AGEs are produced through a process known as the Maillard reaction—reducing sugars react non enzymatically with an amino group of proteins, lipids, and nucleic acids through a series of responses forming a Schiff base, followed by an Amadori rearrangement and subsequent oxidative modifications (glycoxidation). The glycation process is complex and ordinarily slow, but it depends on the availability of nutrients. However, due to the availability of glucose, it is accelerated during hyperglycemia. AGE accumulation can be achieved through either an endogenous or an exogenous source. In the Western diet, for instance, there is a high level of endogenous glucose. However, since cigarette smoke contains highly reactive glycation products, it can also act as a precursor to forming these compounds. In addition, the thermal processing of food, which involves using dry-heat technology, can lead to the formation of substantial amounts of AGE [38].

Studies indicated an essential role for the diacylglycerol (DAG)–protein kinase C (PKC) pathway in regulating AGE production. The PKC system is ubiquitous in cells and regulates various growth factors. It can also generate signal transduction responses to growth factors [39]. PKC may have complex adverse effects on vascular function, including activation of superoxide-producing enzymes, such as nicotinamide adenine dinucleotide phosphate (NADPH) oxidase, and increased expression of uncoupled superoxide-producing endothelial nitric oxide synthase (NOS III). In addition, PKC-mediated superoxide production may inactivate endothelial NOS III-derived NO and may inhibit the activity and/or expression of the downstream NO target, soluble guanylyl cyclase [13]. It was demonstrated that PKC activation increases arginase expression and activity and promotes NOS III phosphorylation, resulting in decreased NO production [40].

The activation of the PKC system can also increase the expression of the growth factor TGF-β, which is a major factor regulating the development of the extracellular matrix. It can also decrease the synthesis of proteolytic enzymes that degrade matrix proteins. The increased expression of the growth factor TGF-β can also lead to the development of structural amyloid in the basement membrane, one of the earliest structural abnormalities in diabetes. The effects of the PKC system on the regulation of fibronectin and type IV collagen are also detrimental in DM [39].

The effects of AGEs on vessel wall homeostasis may explain the rapidly progressive atherosclerosis associated with diabetes. Driven by hyperglycemia and oxidative stress, AGEs are formed to a very rapid degree in diabetes. In the vessel wall, collagen-related AGEs can "capture" plasma proteins, quench NO activity, and interact with specific receptors to modulate many cellular properties [28].

The interaction of AGEs with endothelial cells and other cells that accumulate in the atherosclerotic plaque, such as mononuclear phagocytes and smooth muscle cells, provides a mechanism for increased vascular dysfunction. Specifically, the interaction of AGEs with the vessel wall component increases vascular permeability, the expression of pro-coagulant activity, and the generation of oxidative stress, resulting in increased endothelial expression of endothelial leukocyte adhesion molecules [28].

Endothelium-dependent vasodilation can be evaluated in the coronary and peripheral circulation. Doppler echocardiography, phase-contrast magnetic resonance imaging, and positron emission tomography are the main non-invasive tools to assess coronary endothelial function in the coronary circulation. In any case, the 'gold standard' test requires invasive coronary angiography [41].

The association among DM and endothelial dysfunction is evident in patients with type 1, presenting early (microalbuminuria) or late (macroalbuminuria) nephropathy [42]. The relationship between diabetes and the development of vascular dysfunction in type 1 DM patients is pointed by many markers which correspond to endothelial dysfunction, inclusive EC-dependent low vasodilation, elevated von Willebrand factor (vWF) levels, selectin, thrombomodulin, PAI-1, t-PA, and type IV collagen, as have been demonstrated in this patient population. Endothelial dysfunction is considered an early manifestation of vascular disease in type 2 DM but later in type 1. In addition, studies have presented that cVCAM-1 levels were significantly higher in type 1 DM [10].

Increased calpain activity (calcium-dependent protease) induced in response to hyperglycemia may contribute to DM cardiovascular complications. Studies regarding immunoprecipitation showed that glucose causes NO loss through a calpain-dependent decrease in the association of hsp90 with endothelial NOS. In addition, the inhibition of calpain activity decreased the expression of the endothelial cell surface of the proinflammatory adhesion molecules ICAM-1 and VCAM-1 during hyperglycemia. Moreover, the inhibition of PKC activity reduces leukocyte–endothelium interactions by suppressing the surface expression of endothelial cell adhesion molecules in response to increased oxidative stress [19].

In the last decade, several experimental data have led to the concept that NO is involved not only in the regulation of vasomotor tone but also in vascular homeostasis and neuronal and immunological functions. Endogenous NO is produced by enzymatic conversion of the amino acid l-arginine to l-citrulline by NO-synthase (NOS), of which several isoforms have been recently isolated, cloned, and purified. Neuronal (brain isolate) and endothelial (EC isolate) NOS are called 'NOS-constituents' and produce picomolar levels of NO, of which only a tiny part cause physiological responses. NO produced by NOS type III in the endothelium diffuses to the vascular smooth muscle (VSM), where the enzyme guanylate cyclase is activated, and the concomitant increase in cyclic GMP then induces VSM relaxation [43].

EC produces vasoconstriction-inducing mediators, including endothelin, prostaglandins, and angiotensin II (ANG-II), and regulates vascular tone by maintaining a balance between vasodilation (NO production) and vasoconstriction (e.g., generation A-II). EC produces ANG-II in local tissues and exerts regulatory effects on several VSMC functional activities, including contraction (i.e., vasoconstriction), growth, proliferation, and differentiation [13]. Local concentrations of bradykinin also regulate NOS. This peptide acts with bradykinin B2 receptors on the EC cell surface membrane, increasing NO generation by activating NOS. Significantly, local bradykinin levels are regulated by the angiotensin-converting enzyme (ACE) activity, which breaks down bradykinin into inactive components. In addition, EC has a prominent role in controlling blood flow and restoring the integrity of the vessel wall, preventing bleeding, having an essential function in the balance between coagulation and fibrinolytic components [43].

Among the various factors related to endothelial dysfunction, the gastric peptide ghrelin improves endothelial function in metabolic syndrome. Ghrelin circulates in acylated (AG) and non-acylated (UAG) forms. AG is commonly considered the active form, reduces insulin sensitivity, and exerts orexigenic activity, while more abundant UAG has been considered inactive but has now been shown to exert more biological activity [15].

High glucose indirectly influences endothelial cells by synthesizing growth factors in adjacent cells, significantly transforming growth factor beta (TGFβ) and vascular endothelial growth factor (VEGF), which act on the coagulation cascade, generate thrombin-a potent pro-coagulant cell activator-and fibrin, which can also have stimulating synergic effects on endothelial function [8].

High levels of serum endothelin and E-selectin have been attributed to increased vasoconstrictor properties and increased permeability and leukocyte adhesion. On the other hand, hyperglycemia has been shown to induce overexpression of several matrix metalloproteinases, especially MMP-9, that determine matrix degradation, plaque instability, and atherogenesis [44].

## 5. Mitochondrial Impairment Related to Endothelial Dysfunction in DM

Diabetes is closely associated with endothelial mitochondrial dysfunction, demonstrated by increased oxidative stress, decreased biogenesis, increased DNA damage, and weakened mitochondrial autophagy (mitophagy). Structural (morphological) and functional changes in mitochondria are involved in diabetic endothelial dysfunction [38]. For example, endothelial cell death caused by apoptosis, hyperglycemia, fission, or excessive mitochondrial oxidative stress, are all intervenient in diabetic pathophysiology connected with endothelial dysfunction [45]. High glucose levels have been shown to disrupt mitochondrial dynamics and endothelial cell morphology. In addition, mitochondria linked to the endoplasmic reticulum (ER) are markedly raised in diabetic cardiomyopathy, which may help understand the role of endothelial cell mitochondria in DM complications [45].

Mitochondrial biogenesis is also essential for normal cellular physiological activities. Disorders related to mitochondrial biogenesis result in mitochondrial dysfunction as well. The peroxisome proliferator-activated receptor $1\alpha$ (PGC-$1\alpha$) coactivator is an essential controller of mitochondrial biogenesis, regulated by the SIRT1 factor. Nuclear respiratory factor (NRF)-1 and NRF-2 participate in mitochondrial biogenesis by binding to mitochondrial transcription factor A (mtTFA), which subsequently induces mtDNA transcription and replication [45].

MtDNA damage is frequent in atherosclerotic plaque. MtDNA damage is also commonly seen in patients with metabolic disorders, such as DM, low-glucose tolerance, and metabolic syndrome. In addition, when combined with atherosclerosis, mtDNA damage is intensified in diabetic patients. For example, DM can cause mtDNA base mismatches in retinal endothelial cells. D2 diabetic mice also have increased mtDNA damage, mainly in glomerular endothelial cells [45].

MtDNA of diabetic mice activates endothelial nucleotide-binding oligomerization domain (NOD)-like receptor pyrin domain containing 3 (NLRP3) inflammasomes by mechanisms involving $Ca^{2+}$ influx and the generation of mitochondrial ROS, reduces endothelium-dependent vasodilation, and causes vascular dysfunction. These findings suggest that mtDNA damage affects diabetic vascular complications [38].

Another facet of mitochondrial regulation, which can also be seen in endothelial dysfunction, is mitophagy (autophagy)—mitochondria damaged by numerous stimuli are specifically encapsulated in autophagosomes, then fused to lysosomes, and finally de-graded [12].

In short, mitochondria play a significant role in diabetic vascular complications. Therefore, medications aimed at mitochondrial oxidative stress and dynamics can potentially improve endothelial cell dysfunction, thus, blocking the progression of diabetic vascular complications [45].

## 6. Oxidative Stress Related to Endothelial Dysfunction in DM

Elevated oxidative stress (OS) induced by hyperglycemia aggravates endothelial dysfunction and plays a key role in vascular complications. Excessive oxidative stress results from a disproportionate increase in the production of reactive oxygen species and/or a decrease in the body's total antioxidant capacity [16]. The mitochondrial electron transport chain is the primary source of ROS, which consists of three forms of free radicals: superoxide anion ($O_2^-$), hydroxyl radical (OH), and hydrogen peroxide ($H_2O_2$). Additionally, it produces peroxynitrite, one of the most aggressive toxic free radicals produced starting from nitric oxide through the catalysis of NOS—a reaction usually boosted by excess intracellular calcium. $O_2^-$ is generated by an electron reduction in $O_2$ through cytosolic

cofactors, including NADPH oxidases (NOX) and electron transport chain complexes I, II, and III. $O_2^-$ is converted to $H_2O_2$ by superoxide dismutase 1/2 (SOD 1/2). Excessive accumulation of H2O2 will be converted to OH by metal cations ($Fe^{2+}$ reduced from $Fe^{3+}$ by Haber–Weiss reaction/Fenton Camessy superoxides and Cu+) [45].

Usually, the levels of ROS generated by oxidative phosphorylation are well controlled by the antioxidant system, consisting of glutathione, superoxide dismutase, nicotinamide adenine dinucleotide phosphate (NADPH), and other antioxidants that act as primary regulators of this process. However, under hyperglycemia, pyruvate produced from the breakdown of excessive glucose increases the entry of NADH and FADH2 into the electron transport chain and further increases the voltage gradient along the inner mitochondrial membrane. Hyperglycemia inhibits the activation of the antioxidant system of endothelial cells and has a role in the process of diabetic vascular injury [46].

ROS can cause irreversible oxidative changes in DNA, proteins, and lipids but also play an indispensable role in the regulation of intracellular signaling pathways, such as mitogen-activated protein kinases, redox-sensitive transcription factors (e.g., NF-κB), hypoxia-inducible factors (HIF-1α) [47], and activating protein 1 (AP-1), leading to changes in the expression of downstream target genes or effecter molecules [48].

Abnormal endothelium-dependent vasodilation in patients with DM is due, at least in part, to excessive ROS generated mainly by positive NOX regulation and endothelial nitric oxide synthase (eNOS) decoupling. Significantly increased vascular $O_2$ production and expression of NOX subunits were observed in diabetic subjects, implying that hyperglycemia increases NOX activity. An excess of $O_2^-$ also oxidizes tetrahydrobiopterin (BH4), a cofactor that strictly controls NO production, to increase eNOS decoupling and reduce NO production. Increased ROS and decreased NO will cause irreversible damage to vascular endothelial cells, such as apoptotic damage. NO bioavailability will also be low under high-glucose conditions [45].

It has recently been discovered that accelerated atherosclerosis is associated with insulin resistance and endothelial dysfunction. This precedes the onset of angiopathy and appears to play a central role in the pathogenesis of atherosclerosis in diabetic patients. Indirect support for OS involvement has been provided by restoring endothelial dysfunction in DM cases with vitamin C, a water-soluble antioxidant, and deferoxamine, an iron chelator that prevents the generation of iron-catalyzed hydroxyl radicals. In addition, the ubiquitous and extracellular antioxidant glutathione (GSH-sulfhidrildisufide) has been shown to neutralize oxidants by converting them to other oxidized forms or even to water. In DM, GSH is primarily consumed mainly due to the regeneration of vitamin C, which is strongly oxidized in such patients [49].

The vascular endothelium controls the passage of macromolecules and circulating cells from the blood to the tissues and is a significant target of OS, playing an important role in the pathophysiology of many vascular disorders. Particularly, OS increases vascular endothelial permeability and promotes leukocyte adhesion (inflammatory dimension, see below) coupled with changes in redox-regulated endothelial signal transduction and transcription factors. Furthermore, decreased endothelium-dependent vasodilation in diabetic patients has been associated with impaired NO action secondary to its inactivation resulting from increased OS rather than reduced vascular endothelial NO production [50].

## 7. Immune System Involvement in DM Endothelial Dysfunction

Inflammation is a causative factor of angiopathies [51]. The hallmarks of inflammation are dysfunction and tissue damage, usually manifested by loss of function, heat, pain, and swelling [52]. Pro-inflammatory cytokines activate myeloperoxidase enzymes and NADPH oxidase (NOX), which determine amino acids' oxidation and produce advanced glycation product (AGE) precursors. IL-6, TNF-α, IL-2, IL-1β, interferon-γ (IFN-γ), and MCP-1 are pro-inflammatory cytokines that, together with CRP, are markers of low-grade systemic inflammation (LGI) and prognostic indicators of vascular diseases [52].

Chronic inflammation is linked with low-soluble AGE receptor (RAGEs) levels. It has been suggested that soluble receptors of AGEs ameliorate the pathological condition. Low levels of circulating soluble RAGEs in type 2 DM patients are correlated with resistance to insulin, HbA1c, and CRP, which makes them appropriate markers and predictors of inflammation and cardiovascular events [52].

High levels of other pro-inflammatory cytokines, such as IL-18, are also known to be associated with the development of DM. IL-18 changes have been correlated with changes in insulin resistance, and some studies have reported alterations in IL-18 levels influenced by lifestyle adjustments, such as exercise. The mechanism by which physical activity decreases IL-18 levels is likely by altering insulin signaling or by down-regulating cytokine production, reducing the infiltration of inflammatory cells into adipose tissue. In contrast, serum levels of anti-inflammatory cytokine IL-10 secreted by Th2 subtype T cells, macrophages, and monocytes in humans reduce the risk of cardiac events [52].

Increased levels of pro-inflammatory cytokines, such as IL-6 and TNF-$\alpha$, are known to trigger the development of vascular disorders. Conversely, the actions of IL-10 help promote a reduced risk of inflammation induced in diabetes-related angiopathies. Regular aerobic exercise, medications that decrease proinflammatory activity, a diet high in prebiotic fiber, and cytokines with anti-inflammatory properties have been suggested to mitigate the risk of increased inflammatory processes [52].

AGEs induced release of cytokines will be accompanied by overproduction of various growth factors, such as PDGF (platelet-derived growth factor), IGF-1 (insulin-like growth factor-1), GMCSF (granulocyte/monocyte colony-stimulating factor), and TGF-$\beta$ (transforming growth factor-$\beta$), which affect the function of blood vessel cells. Also, the formation of the immune complexes, which contain modified lipoproteins, is increased. A high level of the immune complex containing altered LDL will raise the risk of macrovascular complications in patients with either type 1 or type 2 DM. The immune complex not only stimulates many cytokine releases but also augments the expression and release of metalloproteinase-1. In addition, activating macrophages by such an immune complex will trigger the release of TNF$\alpha$, which causes up-regulation of CRP synthesis [53].

Macrophages are known to play a role in developing and maintaining host defense and inflammation. They can also perform other homeostatic functions, such as tissue repair and wound healing. The pleiotropic effects of these conditions are mediated by the response of macrophages to environmental signals. These changes can lead to different functional phenotypes, pro-inflammatory or anti-inflammatory. The M1 and M2 polarizations are known to play a role in developing LGI, leading to various complications, such as microvascular and macrovascular diabetes. One of the critical factors that regulate the interaction between macrophages and their environment is the presence of eNO. Although its role in regulating inflammation has not been thoroughly studied, it is believed that this protein could potentially be utilized in treating hyperglycemic conditions [24].

Activation of T cells will also inhibit smooth muscle cells, the proliferation of blood vessel cells, and the biosynthesis of collagen, which will cause vulnerable plaques, causing acute cardiovascular complications/events. In a serial examination of a coronary artery of a patient with type 2 DM after a sudden death, there is an area of necrosis, calcification, and an extensive plaque rupture. In contrast, in type 1 DM, there is an increase in connective tissue content with a few foam cells between the plaques, facilitating a relatively more stable atherosclerosis lesion [18].

## 8. Molecular Pharmacological Approaches in DM Endothelial Dysfunction

Monotherapy is the most common type of first-line therapy for people with diabetes. It is followed by sulfonylureas and dipeptidyl peptidase-4 inhibitor monotherapy [23]. Several therapeutic interventions have been proposed to improve endothelial function in patients with DM (Figure 3). Insulin sensitizers may have a short-term beneficial effect, but the virtual absence of cardiovascular studies precludes any definitive conclusion [35].

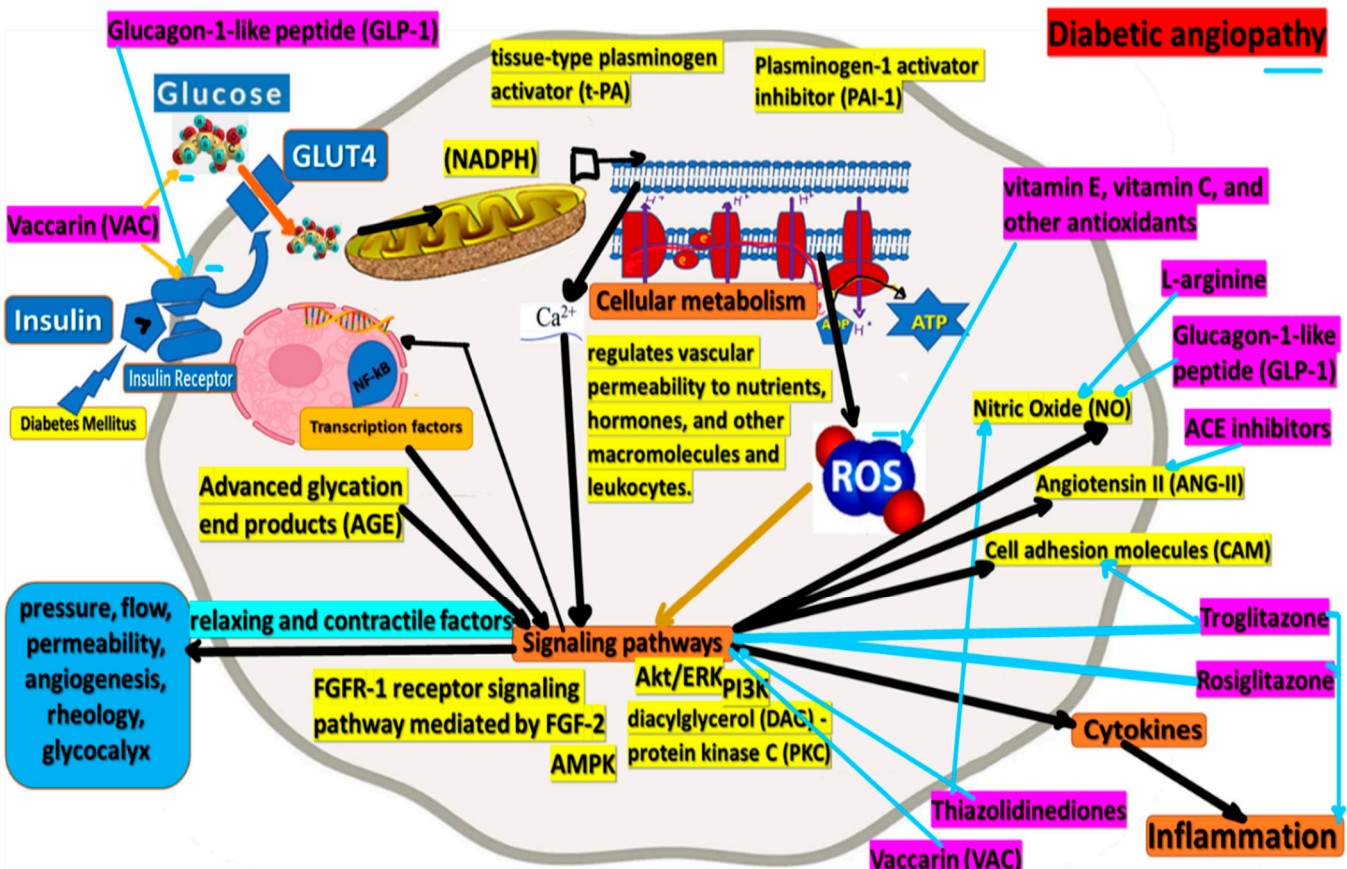

**Figure 3.** Molecular approaches of therapeutic interventions proposed to improve endothelial function in patients with DM.

Rapamycin, commonly used as an antitumoral and immunosuppressive drug, has been shown to effectively prevent aging. It is believed that this drug targets the mTOR gene and the serine/threonine kinase known as FK506. This protein regulates various cellular functions, such as cell growth, mobility, and protein synthesis. Studies have shown that rapamycin can act as a preventive agent by preventing various chronic diseases, such as atherosclerosis, hypertonia, and hypercoagulation. It can also help prevent cancer, Alzheimer's disease, and diabetes.

According to the literature, biguanides, commonly used as a treatment for type 2 diabetes, can also affect the development of geroprotective effects. In addition, these drugs are known to inhibit the growth of gluconeogenesis and increase the sensitivity to insulin.

2 deoxyglucose, a glucose analog, is known to decrease cellular energy metabolism. This substance can also mimic the effects of calorie restriction. Currently, there are not enough data regarding the effects of this substance on the development of agglutination and other cellular processes.

Various natural compounds, such as -tocopherol, ascorbic acid, and ubiquinone, can also help prevent aging. They are believed to be involved in the development of the free radical theory's gerossuppresion.

Since the discovery of resveratrol in 2003, thousands of studies have been conducted on this substance. Data indicated that the drug prevents the development of cataracts and decreases the aorta's elasticity. It is believed that the effects of resveratrol on the growth of senescent cells can be achieved by activating the genes that are known to be involved in the regulation of cellular signaling. These include the p53 and AKT. These genes are also known to influence the expression of other cellular signaling pathways. The researchers explained that the level of this protective factor was increased by the presence of resveratrol on manganese superoxide dismutase. This group of enzymes is

known to degrade the superoxide molecules produced by metabolic processes. In vitro studies conducted on the effects of resveratrol and other polyphenols revealed that they can protect the cardiovascular system by reducing the levels of specific cellular signaling molecules. These include the cytokines that are known to be involved in the regulation of cardiomyocyte apoptosis. It is also known that the presence of this substance can decrease the activity of a cyclo-oxygenase, which is a key component in the synthesis of certain cytokines.

The theory that using antidiabetic substances, such as resveratrol, can improve the function of peripheral nerve and microcirculation tissues is based on the activation of SIRT1. The effects of resveratrol on the development of photo-aging cells were shown to be related to the activation of various cellular signaling pathways. However, despite the multiple advantages of this substance, its effects on the development of human health are still limited due to its low solubility and stability.

Hyperlipidemia and elevated oxidized LDL are risk factors for developing endothelial dysfunction in patients with diabetes. Statins have been widely used to treat hypercholesterolemia in patients with type 2 DM, but there is no evidence to indicate their effect on endothelial function [54].

Hormone replacement therapy using estrogen is thought to improve endothelial function. However, no specific study has been conducted to date to recognize the effect of estrogen penury on endothelial function in DM postmenopausal women [13]. The last half of the 20th century was marked by the approval of estrogen for treating postmenopausal women. It is known that this hormone can prevent various chronic diseases, such as cardiovascular diseases. Unfortunately, the enthusiasm surrounding the use of estrogen has diminished following the publication of studies that indicated the possible side effects of this hormone, such as breast cancer and cardiovascular diseases. In this context, it is essential to note that the effects of estrogen derived from other compounds, such as those from the peloids composition, do not reach the mentioned risks.

$H_2S$ is also known to exert various pharmacological effects, such as its ability to improve the vascular function of the ECs and prevent the development of various complications, also influencing the NO level. In studies on animal models of diabetes, $H_2S$ has been shown to limit the growth of multiple complications, such as kidney disease and retinopathy. In addition, a lack of $H_2S$ homeostasis can also be a contributing factor in the development of hyperglycemic conditions. Due to the increasing number of $H_2S$ donors or exogenous sources discovered in natural sources, including in sulfurous waters used in balneotherapy, researchers are now focusing on developing effective treatment strategies for various diseases, including diabetes and endothelial dysfunction. One of these is the use of organosulfur compounds, which can protect against the development of cardioprotective effects in type 2 diabetes [50]. In a study conducted on animal models of diabetes, Baluchnejadmojarad and colleagues analyzed the properties of S-allyl cysteine, which is the most abundant organosulfur compound in garlic extract. They focused on the potential of this compound to improve the learning and memory deficits in diabetic rats. They also studied the role of the Nrf2, the nuclear factor-kappa B, and the HO-1 signaling in these conditions. The researchers noted that using these compounds could prevent the development of various complications, such as kidney disease and retinopathy. Based on the findings, it has been proposed that using $H_2S$-releasing agents could be a potential treatment option for diabetic patients [50].

L-arginine is a mutual substrate for arginase and NOS [55]. Therefore, it is assumed that L-arginine supplementation may activate NOS, increase NO production, and improve vasodilatation [56]. This hypothesis has been studied in various conditions associated with endothelial dysfunction, such as chronic heart failure, cyclosporine-induced endothelial damage, and type 2 DM. NOS and arginase are substrate-driven enzymes, though arginase is much more active than NOS, and under conditions of mutual substrate deficiency, the NO pathway is inhibited. Accordingly, arginase inhibition increases L-arginine levels, improves NO content, and improves endothelial function [57]. Some researchers have

suggested that vitamin E, vitamin C, and other antioxidant therapies can improve blood vessel function in a diabetic patient [49].

Vaccarin (VAC), an active flavonoid glycoside extracted from the semen of Vaccariae, has extensive biological activities, including protective effects on vascular endothelial cells. VAC attenuates blood glucose, increases glucose and insulin tolerance, reduces lipid metabolism and oxidative stress, and improves endothelium-dependent vasorelaxation in mice STZ/HFD-induced type 2 DM [58].

Glucagon-1-like peptide (GLP-1) is widely used as a drug to treat type 2 diabetes. Exogenous administration of GLP-1 has been shown to reduce blood glucose in patients with type 1 and type 2 diabetes. In addition, GLP-1 can improve insulin resistance and glucose utilization in patients with type 2 diabetes. Furthermore, GLP-1 improves endothelial dysfunction in patients with type 2 DM with established coronary heart disease without affecting the glucose absorption of the whole body. GLP-1 may have increased activation of NO endothelial synthase, thereby interfering with endothelial function [59].

The enzyme dipeptidyl peptidase-4 (DPP-4) is a key component in regulating the glucagon-like peptide-1 (GLP-1) type 2 diabetes. Since this drug is a novel treatment for this condition, the strategy to prevent its inactivation has been studied. A novel series of dihydropyrimidine phthalimide hybrids was synthesized, characterized, and evaluated for their DPP-4 inhibitory activity, as well as antioxidant activity [60]. These compounds increase the levels of active GLP-1 and decrease the secretion of glucose [61]. Examples of DPP-4 inhibitors are Sitagliptin, Vildagliptin, Saxagliptin, Linagliptin, Alogliptin, Gemigliptin, Anagliptin, Teneligliptin, Trelagliptin, Omarigliptin, and Evogliptin [62].

One of the most common drugs used for the treatment of diabetes is teneligliptin. It is a long-acting inhibitor of DPP-4 that can be safely and effectively used in patients with renal impairment. Its unique pharmacokinetics and pharmacologic properties have been shown to provide a wide range of benefits [24].

In animal models, teneligliptin was shown to be effective at preventing the development of atherosclerosis. It also inhibited the growth of atherosclerotic lesions. In a rat model of metabolic syndrome, teneligliptin treatment improved the expression of eNOS. This finding supports the idea that teneligliptin can be used as a non-glycemia-based therapy for this condition. Previous studies showed that the effects of multiple DPP-4 inhibitors on the endothelium were beneficial. However, in these studies, it was also revealed that teneligliptin had antioxidant properties. This drug was able to reduce the levels of certain inflammatory markers and activate the transcriptional cascades of antioxidant genes at the cellular level [24].

Although there are currently no single treatment options for type 2 diabetes, the two most common drugs used for type 2 DM are sitagliptin and vildagliptin. These two drugs can be used once daily, and they have demonstrated a reduction in the levels of A1C by 1%. In drug-naive patients, the effects of these two drugs can be observed for up to 52 weeks. In combination with other drugs, such as metformin, these two drugs can improve glycemic control and lower the levels of A1C. They are safe and are both body-weight neutral. The studies that have been conducted so far suggest that the use of DPP-4 inhibitors as a monotherapy or combination therapy is an effective strategy for type 2 diabetes [63].

Although ACE inhibitors can improve the function of the blood vessels, their effects on endothelial cells are still unclear. Notably, it was shown that ACE inhibitors, such as ramipril and enalapril, and angiotensin II (AT II) inhibitors, such as valsartan, increased endothelial progenitor cell levels in patients, probably interfering with the CD26/DPP-4 system [64]. Despite the various risks associated with these substances, ACE inhibitors consistently prevent coronary artery disease, stroke, and diabetic microvascular complications of nephropathy and retinopathy. In addition, inhibition of the renin-angiotensin system is associated with a reduced incidence of DM [54]. On the other hand, Angiotensin II expresses pro-oxidative effects on the vascular system. It can decrease NO bioavailability and lead to vascular damage [64].

Peroxisome-proliferator-activated receptor (PPAR) ligands, troglitazone, and rosiglitazone, also improve endothelial function. In addition, several studies have shown improvements in the reactivity of the brachial artery in patients with DM [5].

Troglitazone in vivo inhibits the expression of VCAM-1 and ICAM-1 on activated endothelial cells. This medicine also decreases the amount/content of monocytes/macrophages in the atherosclerotic plaque. The drug reduces the expression of VCAM-1, ICAM-1, and E-selectin, which are induced by oxidized LDL and TNFα. Troglitazone may improve insulin sensitivity [13].

Rosiglitazone improves coronary artery endothelial function in patients with insulin resistance who do not have traditional risk factors for atherosclerosis, as well as impaired glucose or diabetes tolerance [54].

Thiazolidinediones improve insulin-mediated glucose uptake in target insulin tissues by activating the PPAR. They directly affect adipose tissue by suppressing TNFα and possibly leptin expression, suppressing lipolysis. Thus, they decrease plasma concentrations of free fatty acids and increase plasma adiponectin levels and directly affect insulin-mediated glucose transport in the skeletal muscle and heart. Thiazolidinedione administration reverses insulin resistance and many components of metabolic syndrome. Treatment is generally associated with increased HDL cholesterol levels, blood pressure, plasma triglyceride levels, small LDL cholesterol, PAI-1 levels, and albumin excretion rates, and lower glucose and hemoglobin A1C [54]. With an increase in insulin sensitivity and a decrease in fasting insulin and free fatty acid levels, thiazolidinediones in combination with hormone therapy (HT) in postmenopausal women were shown to reduce endothelial function [65].

The mechanisms by which thiazolidinediones improve endothelium-dependent blood flow are unknown, but they probably involve several effects. First, as described above, they have significant anti-inflammatory effects involving a decrease in circulating adipokines (e.g., TNFα, PAI-1, leptin), which is reflected by low levels of high-sensitivity CRP; increase in adiponectin levels; and decreased vascular expression of adhesion molecules. Secondly, insulin is a vasodilator that stimulates eNOS expression via phosphatidylinositol 3-kinase (PI3K). This effect of insulin is attenuated in patients with insulin resistance. Additionally, PPAR is expressed in endothelial cells, and its ligands are reported to improve NO production, possibly by stimulating the PI3K pathway and, therefore, eNOS expression. Thirdly, PPAR ligands improve some components in metabolic syndrome that could adversely affect endothelial function, involving low levels of HDL cholesterol, high levels of triglycerides and free fatty acids, high blood pressure, and carbohydrate intolerance. PPAR ligands also decrease oxidative stress and, thus, can improve the vascular balance between NO and vasoconstrictors. Finally, an endogenous NOS inhibitor is associated with reduced NO-mediated vasodilation and increased adhesion of mononuclear cells to the endothelium [54].

In normal conditions, all glucose is absorbed through the kidney tubules [66]. The early S1 segment of the kidney is responsible for the reabsorption of 80% to 90% of the glucose, while the S2 and S3 sections are responsible for the remaining 10% to 20%. The glucose that escapes from the soluble glucose receptor 2 (SGLT2) is then reabsorbed by the glucose-carrying protein known as SGT1 [67]. Studies show that people with DM have an elevated level of the renal threshold for glycosuria, and their reabsorption capacity is also increased. This is due to the presence of certain types of cells that are known to increase the expression of glucose-binding protein 2 (SGLT2) [67]. This mechanism is believed to contribute to the development of hyperglycemia in these individuals. In 1835, the drug phlorizin was extracted from apple trees. It is the first natural inhibitor of the two major types of glucose-binding protein 2 that has a high-affinity and competitive activity. Initially, it was used for the treatment of various infectious diseases, such as malaria and fever. However, it was later discovered that this drug can cause glycosuria. Various pharmaceutical companies have been developing compounds that are capable of inhibiting the activity of glucose-binding protein 2 using different methods. In 2008,

Dapa was developed, which has a higher level of potency than that of the drug known as SGT1. The second type of phlorizin derivative, called Cana, is also known to have higher inhibitory activity against this protein.

The third type of phlorizin derivative that is capable of inhibiting the activity of this protein is Empa. It has a high selectivity over the commercial version of this drug, known as SGT1. The development of the new generation of antidiabetic agents, known as SGLT2i, was carried out through a series of phase III studies. These studies were designed to examine the effects of these drugs on the glycemic levels of patients. Empa is a third-generation SGLT2i that is available in the US and Europe. In various phase III studies, it has been shown that it can lower the levels of glucose in the blood of people with type 2 diabetes. After 24 weeks of treatment, the researchers found that the drug reduced the levels of glucose in the blood of these patients by 0.74% and 0.85%. The most recent addition to the treatment of type 2 diabetes, Ertu, is known to reduce the levels of glucose in the blood of patients with inadequately controlled blood glucose levels. After 26 weeks of treatment, the researchers found that the drug reduced the levels of HbA1c by almost 1%. The results of the study, which were maintained for 52 weeks, were also favorable [68].

## 9. Discussion

Although diabetes is known to increase the risk of heart disease and stroke, its effects on different sexes, ages, and levels of conventional risk factors are not well understood. For instance, the extent to which this condition is associated with fatal or non-fatal myocardial infarction or hemorrhagic stroke is not well explained. Further, how much of the impact of diabetes on vascular pathology can also be associated with factors such as blood pressure, cholesterol, and obesity? This condition is also known to confer a two-fold excess risk of death due to other vascular causes. In people without diabetes, the effects of dysglycemia on their vascular health are not well understood. Although fasting blood glucose is known to be associated with an increased risk of vascular disease, the exact mechanisms by which this condition affects this parameter are not well known [29].

In type 1 DM, increased blood flow occurs in tissues prone to vascular complications, such as the kidney or retina. This is correlated with a glucose-mediated increase in endothelial formation [69]. Retinopathy and nephropathy are manifestations of diabetic microangiopathy in relatively young patients under poor glycemic control. There is a possibility of endothelial dysfunction in type I diabetes, over time, by increased vasodilation based on increased formation of the endothelial-derived relaxation factor at the beginning of the disease, followed by "pseudonormalization" of endothelial function; finally, it results in a reduction in endothelial-dependent relaxation and the morphological manifestation of angiopathy [70]. The mechanisms that mediate this chronology of endothelial dysfunction remain to be investigated, but endothelial defects in the L-arginine metabolism may be involved, as well as reduced smooth muscle cell function, as seen in hypercholesterolemia.

Mitochondrial encephalomyopathy syndrome lactic acidosis and stroke (MELAS) [71] (clinically characterized by recurrent headaches, hearing loss, diabetes, and short stature). DM develops into MELAS syndrome due to multiple defects in insulin and glucose metabolism, including insulin deficiency, increased gluconeogenesis, and insulin resistance in pancreatic β-cells; an ATP-sensitive potassium channel is required for insulin release. Decreased ATP synthesis due to mitochondrial dysfunction may lead to impaired insulin secretion and insulinopenia. In addition, two factors may contribute to the increase in gluconeogenesis in MELAS syndrome: insulin resistance, which results in reduced insulin inhibition of hepatic gluconeogenesis, and lactic acidemia, which may fuel hepatic gluconeogenesis [71].

DM is also a risk factor associated with severe complications in SARS-CoV-2 infection, which causes hyperglycemic emergencies, such as hyperosmolar hyperglycemic status (HHS) and diabetic ketoacidosis (DKA) [72]. In addition, DM can cause a diseased immune system by predisposing it to infection. The endothelial receptor of the angiotensin 2 converting enzyme (ACE2) responsible for SARS-CoV-2 invasion in human cells has reduced

expression in patients with diabetes, possibly due to glycosylation. In the case of COVID-19 patients, there is a high incidence of coagulative disorders, generating a disseminated intravascular coagulation, which is lethal in severe conditions. $H_2S$ is, therefore, of therapeutic interest, interfering with both vascular inflammation and coagulation [12]. Insulin resistance and impaired glucose homeostasis have been shown to cause alveolar capillary microangiopathy and interstitial fibrosis due to super inflammation [72]. In addition, elevated HbA1c levels have been associated with inflammation and hypercoagulability, leading to an increased mortality rate in patients with diabetes with COVID-19 [73].

Lung histopathological analysis found multiple thrombi in the small to medium pulmonary arteries, giving rise to the theory of immunothrombosis, associated with COVID-19, contrary to the conventional thromboembolic mechanism of PE. In situ microvascular thrombosis or immunothrombosis occurs due to alveolar injury, inflammatory storm, and disruption in the thromboprotective pulmonary vascular endothelium. The clinical results of COVID-19 are worse in patients with diseases associated with endothelial dysfunction, such as systemic hypertension, diabetes, and obesity [73].

One of the treatment modalities presented in the previous section is also $H_2S$ present, including in sulfurous waters and therapeutic sapropelic mud used in balneotherapy. Regarding this therapeutic strategy, we must discuss that usually, hydrotherapy is contraindicated for diabetes and endothelial dysfunction due to the vascular changes involved in such a treatment modality, which is also connected with water temperature [74].

## 10. Conclusions

The future holds great promise. Despite the various studies on the above-presented drugs, their effects on the pathological process in endothelial dysfunction are still not fully understood (most probably as a result of—as detailed above—very complex and complicated pathophysiology of endothelial dysfunction and its lesional consequences and consecutive complication). Extensive research is needed along with randomized studies in this entity, especially in molecular biology and genetic engineering, to better explore the hidden aspect of the iceberg. Numerous biomarkers of DM microangiopathy are already known and may be therapeutic targets in the future.

The future will show a growing interest in finding reliable methods for testing endothelial function. As endothelial dysfunction measures become clinically applicable, this may result in improved risk assessment methods that would help predict, prevent, and treat cardiovascular-related diseases. Inflammatory markers are likely to find their way into risk assessment; several therapeutic strategies to improve endothelial function in DM are being investigated. Unfortunately, despite substantial progress in understanding how microvascular and macrovascular complications develop, much effort is still needed.

**Author Contributions:** All authors had specific but overall equal contributions in achieving this article: conceptualization, G.O. and C.M.; methodology, G.O., A.A., M.-A.T. and C.M.; validation, M.R., I.A., G.D. and S.C.S.; formal analysis, F.C.F. and E.V.I.; data curation, C.M., A.M.I. and C.O.; writing—original draft preparation, G.O. and C.M., writing—review and editing, F.C.F., C.M. and E.V.I.; visualization, A.A., F.C.F., G.O. and C.M.; supervision, G.O. and C.M. All authors contributed to data analysis, drafting, or revising the article. All authors have read and agreed to the published version of the manuscript.

**Funding:** This research received no external funding.

**Conflicts of Interest:** The authors declare no conflict of interest.

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
