# Peer review of "Topical Reappraisal of Molecular Pharmacological Approaches to Endothelial Dysfunction in Diabetes Mellitus Angiopathy"

_cimb, doi:10.3390/cimb44080233_

Round 1

Reviewer 1 Report

Overall Muntenau et al. put forward a straightforward and well-written manuscript.

 Although it is well written, a few modification should be made:

1.       Page 2 lines 62-64: citation number 11 should be changed. This cited paper does not talk about altered oxidized/reduced forms of NAD+/NADH rations in diabetes.

2.       Page 3, lines 18-123: some references may be missing.

3.       Page 4, line 166-167: this sentence should be rephrased/specified, because as we know microvascular disease not only can affect the small blood vessels in the heart, but can have an effect on multiple organs in the body.

4.       Page 5, line 208-211 a reference may be missing.

5.       Page 16: lines 719-720: ACE was already defined on page 8.

6.       Page 16, lines 728-730: reference is missing.

Author Response

Dear reviewer,

Thank you very much for your evaluation of our article, submitted for CIMB. Your constructive suggestions were included in our manuscript and below are answered point by point:

  1. Page 2 lines 62-64: citation number 11 should be changed. This cited paper does not talk about altered oxidized/reduced forms of NAD+/NADH rations in diabetes. - we changed the previous wrong citation (due to file management) with: Stehouwer CDA, Lambert J, Donker AJM, Van Hinsbergh VWM. Endothelial dysfunction and pathogenesis of diabetic angiopathy. Cardiovasc Res. 1997;34(1):55–68., in which: ”alterations in the cellular redox state by an altered NADH/NAD+ ratio”
  2. Page 3, lines 118-123: some references may be missing. - introduced: ÖZTÜRK Z. Diabetes, Oxidative Stress and Endothelial Dysfunction. Bezmialem Sci. 2019;7(1):52–7. 
  3. Page 4, line 166-167: this sentence should be rephrased/specified, because as we know microvascular disease not only can affect the small blood vessels in the heart, but can have an effect on multiple organs in the body. - we rephrased it as follows: Microvascular disease is a type of vascular disease that can affect the small blood vessels on multiple organs in the body.
  4. Page 5, line 208-211 a reference may be missing. - introduced: Sarwar N, Gao P, Kondapally Seshasai SR, Gobin R, Kaptoge S, Di Angelantonio E, et al. Diabetes mellitus, fasting blood glucose concentration, and risk of vascular disease: A collaborative meta-analysis of 102 prospective studies. Lancet [Internet]. 2010;375(9733):2215–22. Available from: http://dx.doi.org/10.1016/S0140-6736(10)60484-9
  5. Page 16: lines 719-720: ACE was already defined on page 8. - correctly, we used in the new version only the shorted form ACE
  6. Page 16, lines 728-730: reference is missing. -  introduced: Shab-Bidar S, Neyestani TR, Djazayery A, Eshraghian MR, Houshiarrad A, Gharavi A, et al. Regular consumption of vitamin D-fortified yogurt drink (Doogh) improved endothelial biomarkers in subjects with type 2 diabetes: A randomized double-blind clinical trial. BMC Med [Internet]. 2011;9(1):125. Available from: http://www.biomedcentral.com/1741-7015/9/125

Reviewer 2 Report

Summary: In the manuscript entitled "Topical reappraisal of molecular pharmacological approaches to endothelial dysfunction in Diabetes Mellitus angiopathy," Constantin Munteanu et al. interrogate five medical databases to present the state-of-the-art data related to endothelial dysfunction in diabetic micro- and macroangiopathy.

Overall, this is potentially a good report summarizing several sets of highly relevant results. It opens a new avenue to the understanding of etiology and to the treatment of endothelial dysfunction associated with DM.

However, in its current form, this study has several issues that should be carefully addressed.

Critiques:

General:

1.            Introduction.

a.       The authors state (lines 69-71) that "the actions of insulin-stimulating agents, such as reducing vascular endothelium's nitric oxide, can also improve the blood flow and increase glucose uptake in skeletal muscle. The actions of insulin are mainly mediated by the phosphatidylinositol-3-kinase 72 (PI3K) ". However, the authors of the cited article (reference 14) suggest that "vasodilator actions of insulin-stimulating production of nitric oxide (NO) from vascular endothelium increases blood flow that further enhances glucose uptake in skeletal muscle".   Please, clarify.

2.            Diabetes Mellitus – vascular pathological interplay picture

a.       Line 130 b-cells. Please change to β-cells.

3.            Endothelial dysfunction in Diabetes Mellitus

a.       The authors state (line 225-226) that "evidence suggests that endothelial dysfunction is a precursor of diabetic angiopathy." However, here they cite a review paper, which does not provide a new piece of evidence. Moreover, the word precursor is not appropriate here. It is, in fact, associated with angiopathy.

b.      Line 236. Nitric oxide. Use abbreviation. Generally, all abbreviations/acronyms should be written out in full on first use (in both the abstract and the paper itself) and followed by the abbreviated form in parentheses.

c.       Lines 271-273. The authors correctly state that "metabolic disorder in Diabetes is associated with excess ROS generation through polyol, glucose self-oxidation, prostanoid synthesis, and protein glycosylation. Many studies in animal models and human clinical trials strongly suggest that hyperglycemia is a key feature of diabetic microvascular complications". Unfortunately, they do not support this statement with references. Moreover, an additional mechanism that increases ROS generation via activation of polyamine metabolism should be mentioned (1).

d.      Lines 299-301. The authors correctly state that "PKC-mediated superoxide production may inactivate endothelial NOS III-derived NO and may inhibit the activity and/or expression of the downstream NO target, soluble guanylyl cyclase." In fact, the mechanism is more complex. In general, the endothelial metabolism of arginine plays a key role in vascular homeostasis. There is a strong relationship between arginine transport and the activity of arginase. It was demonstrated that PKC activation increases arginase expression and activity and promotes NOS III phosphorylation, resulting in decreased NO production (2).

e.       Lines 347-348. NOS type I (brain isolate) change to neuronal. Type III (EC isolate) change to endothelial.

f.        Line 668. The authors state that "L-arginine is a substrate of NOS formation."

In fact, L-arginine is a mutual substrate for arginase and nitric oxide synthase (NOS) (3). Arginase is a manganese-containing enzyme that converts L-arginine into L-ornithine and urea. NOS isoforms, in turn, catalyze the production of NO and citrulline.

g.      Lines 669-670. The authors correctly mentioned that "it is assumed that L-arginine supplementation may activate NOS, increase NO production, and improve vasodilatation." However, they do not provide a reference. An article by Jessica Gambardella et al. (2020) might be quoted here (4).

h.      Line 672. The authors state that "it is not yet clear how L-arginine levels can increase NOS activation." In fact, NOS and arginase are substrate-driven enzymes. Though arginase is much more active than NOS and under conditions of mutual substrate deficiency, the NO pathway is inhibited. Accordingly, arginase inhibition increases L-arginine levels, improves NO content, and improves endothelial function (5).    

References:

1.      Kulkarni A, Anderson CM, Mirmira RG, Tersey SA. Role of Polyamines and Hypusine in β Cells and Diabetes Pathogenesis. Metabolites. 2022 Apr 12;12(4):344. doi: 10.3390/metabo12040344. PMID: 35448531; PMCID: PMC9028953.

2.      Visigalli R, Barilli A, Parolari A, Sala R, Rotoli BM, Bussolati O, Gazzola GC, Dall'Asta V. Regulation of arginine transport and metabolism by protein kinase Calpha in endothelial cells: stimulation of CAT2 transporters and arginase activity. J Mol Cell Cardiol. 2010 Aug;49(2):260-70. doi: 10.1016/j.yjmcc.2010.04.007. Epub 2010 Apr 26. PMID: 20430034.

3.      Gilinsky MA, Polityko YK, Markel AL, Latysheva TV, Samson AO, Polis B, Naumenko SE. Norvaline Reduces Blood Pressure and Induces Diuresis in Rats with Inherited Stress-Induced Arterial Hypertension. Biomed Res Int. 2020 Feb 12;2020:4935386. doi: 10.1155/2020/4935386. PMID: 32149110; PMCID: PMC7042509.

4.      Gambardella J, Khondkar W, Morelli MB, Wang X, Santulli G, Trimarco V. Arginine and Endothelial Function. Biomedicines. 2020 Aug 6;8(8):277. doi: 10.3390/biomedicines8080277. PMID: 32781796; PMCID: PMC7460461.

5.      Polis B, Gurevich V, Assa M, Samson AO. Norvaline Restores the BBB Integrity in a Mouse Model of Alzheimer's Disease. Int J Mol Sci. 2019 Sep 18;20(18):4616. doi: 10.3390/ijms20184616. PMID: 31540372; PMCID: PMC6770953.

Author Response

Dear reviewer,

Thank you very much for your highly constructive and helpful evaluation of our article, submitted for publication in CIMB. We used your suggestions to improve our article and below are our responses point by point to the received comments and suggestions:

  1. The authors state (lines 69-71) that "the actions of insulin-stimulating agents, such as reducing vascular endothelium's nitric oxide, can also improve the blood flow and increase glucose uptake in skeletal muscle. The actions of insulin are mainly mediated by the phosphatidylinositol-3-kinase 72 (PI3K) ". However, the authors of the cited article (reference 14) suggest that "vasodilator actions of insulin-stimulating production of nitric oxide (NO) from vascular endothelium increases blood flow that further enhances glucose uptake in skeletal muscle".   -  there was a wrong formulation that was corrected: The actions of insulin-stimulating agents, by stimulating the production of vascular endothelium's nitric oxide, can also improve the blood flow and increase glucose uptake in skeletal muscle.
  2. Line 130 b-cells. Please change to β-cells. - corrected
  3. The authors state (line 225-226) that "evidence suggests that endothelial dysfunction is a precursor of diabetic angiopathy." However, here they cite a review paper, which does not provide a new piece of evidence. Moreover, the word precursor is not appropriate here. It is, in fact, associated with angiopathy. we reformulated: The function of the vascular endothelium is altered in DM, conducting to the clinical expression of microangiopathies. Diabetic endothelial dysfunction is thought to be generated by an early increase in vascular permeability.
  4. Line 236. Nitric oxide. Use abbreviation. Generally, all abbreviations/acronyms should be written out in full on first use (in both the abstract and the paper itself) and followed by the abbreviated form in parentheses. - resolved, the abbreviations were checked
  5. Lines 271-273. The authors correctly state that "metabolic disorder in Diabetes is associated with excess ROS generation through polyol, glucose self-oxidation, prostanoid synthesis, and protein glycosylation. Many studies in animal models and human clinical trials strongly suggest that hyperglycemia is a key feature of diabetic microvascular complications". Unfortunately, they do not support this statement with references. Moreover, an additional mechanism that increases ROS generation via activation of polyamine metabolism should be mentioned (1). It was rephrased: Metabolic disorder in diabetes is associated with excess ROS generation through polyol, glucose self-oxidation, prostanoid synthesis, and protein glycosylation (19). An additional mechanism that increases ROS generation via activation of polyamine metabolism, involved in cell survival, cell proliferation, and protein synthesis contributes to diabetes pathogenesis under conditions of inflammation (36).
  6. Lines 299-301. The authors correctly state that "PKC-mediated superoxide production may inactivate endothelial NOS III-derived NO and may inhibit the activity and/or expression of the downstream NO target, soluble guanylyl cyclase." In fact, the mechanism is more complex. In general, the endothelial metabolism of arginine plays a key role in vascular homeostasis. There is a strong relationship between arginine transport and the activity of arginase. It was demonstrated that PKC activation increases arginase expression and activity and promotes NOS III phosphorylation, resulting in decreased NO production (2). - the suggested explanation was introduced and cited after the mentioned paragraph. Thank you for the interesting related reference.
  7. Lines 347-348. NOS type I (brain isolate) change to neuronal. Type III (EC isolate) change to endothelial. - the changes have been made
  8. Line 668. The authors state that "L-arginine is a substrate of NOS formation." In fact, L-arginine is a mutual substrate for arginase and nitric oxide synthase (NOS) (3). Arginase is a manganese-containing enzyme that converts L-arginine into L-ornithine and urea. NOS isoforms, in turn, catalyze the production of NO and citrulline. - the suggested reference was introduced and the text was accordingly changed regarding L-arginine as a mutual substrate for arginase and NOS.
  9. Lines 669-670. The authors correctly mentioned that "it is assumed that L-arginine supplementation may activate NOS, increase NO production, and improve vasodilatation." However, they do not provide a reference. An article by Jessica Gambardella et al. (2020) might be quoted here (4). - the suggested reference was used to sustain the mentioned text. 
  10. Line 672. The authors state that "it is not yet clear how L-arginine levels can increase NOS activation." In fact, NOS and arginase are substrate-driven enzymes. Though arginase is much more active than NOS and under conditions of mutual substrate deficiency, the NO pathway is inhibited. Accordingly, arginase inhibition increases L-arginine levels, improves NO content, and improves endothelial function (5).  - We rephrased the paragraph and included the provided explanation sustained by the proposed reference. Thank you!